1

# 1An innovative STEM outreach model to foster the next generation of2geoscientists, engineers, and technologists (OH-Kids)

Adrián Pedrozo-Acuña<sup>\*1,2</sup>, Roberto J. Favero Junior<sup>1,2</sup>, Alejandra Amaro-Loza<sup>1</sup>, Roberta K. Kurek Mocva<sup>1</sup>,
 Juan A. Sánchez-Peralta<sup>1,2</sup>, Jorge A. Magos-Hernández<sup>1,2</sup>, Jorge Blanco-Figueroa<sup>1</sup>
 <sup>1</sup>Universidad Nacional Autónoma de México, Instituto de Ingeniería, Av. Universidad 3000, Coyoacán, 04510, Mexico City, Mexico.
 <sup>2</sup>Mexican Institute of Water Technology (IMTA), Paseo Cuauhnáhuac 8532, Col. Progreso Jiutepec, Morelos, Mexico.

#### 8 Abstract

7

9 Childhood education programmes aiming at incorporating topics related to science, technology, 10 engineering, and mathematics (STEM) have gained recognition as key levers in the progress of 11 education for all students. Inspiring young people to take part in the discovery and delivery of 12 science is of paramount importance not only for their well-being but also for their future human 13 development. To address this need, an outreach model entitled OH-Kids was designed to empower educators and pupils through the development of high-quality STEM learning 14 15 experiences based on a research project. The model is an opportunity for primary school learners 16 to meet geoscientists while receiving the take-home message that anyone can get involved in 17 scientific activities. The effort is part of a research project aimed at the real-time monitoring of 18 precipitation in Mexico City, which is a smart solution to rainfall monitoring using information 19 and communications technologies. The argument behind this effort is that in order to produce the 20 next generation of problem-solvers, education should ensure that learners develop an appreciation and working familiarity with a real-world project. Results show success at 21 22 introducing the role of researchers and STEM topics to 6-12-year-old learners. 23

- 23
- 24
- 25
- 27

<sup>\*</sup> Corresponding author now at Mexican Institute of Water Technology

Email: apedrozoa@ii.unam.mx

## 28 **1. Introduction**

Inspiring young people to take part in the discovery and delivery of science, technology, 30 engineering, and mathematics (STEM) has been proved to contribute significantly not only to 31 their well-being, but also to their future human development (Bertram and Pascal, 2016; Morgan 32 et al., 2016; Friedman-Krauss et al., 2018). However, the current system uses teaching and learning 33 methods that tend to develop geoscientists, engineers, and technologists only by mere chance. It 34 seems that, in far too many cases, teachers and/or syllabi unintentionally deter potential STEM 35 learners – especially girls – due to the way they choose to teach science, mathematics, design, and 36 technology.

37

Moreover, in several countries, such as the United Kingdom, STEM topics do not appear on the timetables of pupils of primary or lower secondary age (Bianchi and Chippindall, 2018). This is also the case in Mexico, where promoting and improving student engagement on these topics constitutes a great challenge for teachers. This is ascribed to the lack of professional guidance for early childhood educators, who rarely receive in-depth professional training for teaching STEM (Breneman et al., 2009).

44

Primary and secondary education have been found to be significant periods for developing students' interest in science and technology (Maltese et al. 2014). At these stages, pupils' interest in science is closely related to the level of appreciation of its applicability in their lives. Therefore, it is important to incorporate activities in the classroom that convey the wider relevance of science to everyday life (Sheldrake et al., 2017). This may encourage students' aspirations towards science and engineering careers (Regan and DeWitt, 2015).

51

52 On the other hand, scientific and technological education is evolving rapidly with the 53 advancement of the digital age, so it is inevitable for pupils and academics to transit to the 54 development and use of new strategies that allow the overcoming of observed difficulties in the 55 teaching-learning process (Souza et al., 2018). The generation of new strategies of STEM 56 communication to children constitutes a critical step towards improving not only their learning 57 experience, but also their teaching practice.

3

59 It has been acknowledged that promoting the relevance and utility of science to students enhances 60 their interest in science and boosts their attainment (Rozek et al., 2015; DeWitt and Archer, 2015; 61 Savelsbergh et al., 2016; Sheldrake, 2016). In this regard, a good strategy for promoting meaningful 62 learning is the development and implementation of educational activities through the exposure of 63 pupils to a real-world application of STEM (Smith et al., 2005). The argument behind this effort is 64 that, in order to produce the next generation of problem-solvers, education should ensure that 65 learners develop an appreciation of and a working familiarity with STEM disciplines and human-66 environmental systems (Breneman et al., 2018). 67 68 This study presents an innovative teaching experience to enhance positive effects on the attitude 69 of students towards STEM disciplines (Savelsvbergh et al., 2016). The outreach model entitled OH-70 Kids was designed to empower educators and pupils through the development of STEM learning 71 experiences focused on water resources and based on a research project. 72 73 The initiative is part of the real-time Hydrological Observatory of the National Autonomous 74 University of Mexico's Institute of Engineering (OH-IIUNAM), which comprises a research project 75 aimed at the real-time monitoring of precipitation in Mexico City (Pedrozo-Acuña et al. in review). 76 The system represents a smart-water solution comprised by the application of information and 77 communications technologies within an urban environment. Notably, the framework highlights 78 the integration of geoscientists from hydrology, meteorology, and geology, as well as 79 technologists and various practical engineering specialists (hydraulics, electronics). 80

81 The outreach model OH-Kids was born from the interaction of the team behind this research 82 project with primary school educators. This communication resulted from the installation of 54 83 stand-alone stations to measure precipitation in Mexico City, 15 of which were installed on the 84 rooftops of primary schools and nine on the rooftops of secondary schools. In the primary schools, 85 the equipment installation process prompted the interest of educators and pupils in the apparatus 86 and the wider application of science and engineering in the project. This research project was 87 therefore seen as an opportunity to develop an innovative approach focused on improving the 88 attitudes of the students towards science and engineering.

4

Applied in several primary schools and one science fair in Mexico City, the programme has a design strategy based on principles relevant to all researchers working with educators in settings that include families from different social backgrounds. The paper is organised as follows: Section 2 presents a methodological description of the outreach model, Section 3 introduces the results of the implementation of this programme; finally, Section 4 summarises the main conclusions.

95

# 96 **2.** OH-Kids: a *STEM* outreach model

97 The OH-Kids outreach model comprises a collaborative and effective researcher-educator-learner 98 communication strategy-built around principles of engagement and guided discovery 99 learning – to encourage STEM subject uptake. In this sense, the model was designed following an 00 approach that involves the exposure of learners to a real-world-application STEM project using games and hands-on activities as educational and didactic tools (Parson and Miles, 1994; Poljak et 01 .02 al., 2018). Our proposal for the model design was based on different documented efforts that had .03 an effect on classroom practice and child outcomes (Design-Based Research Collective 2003; 04 Drago-Severson 2009; Zaslow et al., 2010).

.05

Following Rogers et al. (1988), the activities are designed on the basis that involvement of children is beneficial for their learning process, using their innate curiosity as a starting point (McIntyre, 1984 and Piaget, 1971). Educational games and activities have been recognized as a valid strategy to enhance student engagement and develop key skills that may be applicable in other contexts, such as the ability to work as a team (Butucha, 2016; Garris et al., 2002). Students learn as a consequence of playing, which promotes meaningful learning (Croxton and Kortemeyer, 2018).

The creation of a learning experience that promotes participation and science teaching in an active learning environment has positive effects on all children regardless of their literacy and social origin. Indeed, this approach has been found to create a rich learning environment that is accessible to all the students in the classroom (Fantuzzo et al., 2011; Sarama et al., 2012). Furthermore, studies have shown that there is a correlation between positive experiences in science as a child and a strong interest in and a positive perception of science as an adult (Falk et al., 2017).

.20

5

| 21 | The i | intention | of th | e OH-Kids | outreach | model | is | to | enhance | thinking | skills | related | to | STE | M |
|----|-------|-----------|-------|-----------|----------|-------|----|----|---------|----------|--------|---------|----|-----|---|
|----|-------|-----------|-------|-----------|----------|-------|----|----|---------|----------|--------|---------|----|-----|---|

- disciplines with a focus on water resources in the classroom, in addition to highlighting the use of science and engineering for everyday life and the wellbeing of society. The OH-Kids model can
- 24 serve as an example to other researchers and teachers interested in childhood STEM education. It
- 25 can also support different learners in other settings, such as museums and science centres.
- 26 The model's main objectives are:
- a) To enable pupils to meet a scientific team and find out what scientists do and how researchis carried out
- b) To improve understanding of hydrology, specifically the water cycle, and the relationshipbetween water and cities
- c) To enhance children's experience of science and technology
- d) To demonstrate links between topics covered in school curricula and research projects inthe real world
- e) To know the extent to which this methodology results in changes in learners' enjoymentand perception of science and scientists.
- .36

The model is akin to a workshop and includes two types of activities. The first one is related to traditional academic strategies: a short lecture that explains the real-world problem and associated concepts. The second one consists of interactive and ludic activities that are implemented within the classroom as didactic tools for teaching the STEM disciplines involved in water resources. All activities are organised in a circuit that enables the alternation of an academic activity with one of a more ludic nature, which can be considered as one of the innovations of this outreach model.

- In order to assess modifications in pupils' perceptions of some basic concepts related to water 44 45 resources, as well as their perceptions of science and scientists, resulting from the application of 46 our model, we designed a diagnosis and final questionnaire. Similar instruments have been 47 documented to establish attitudes within educational research (Muller et al., 2013; OECD, 2016). 48 Tables 1 and 2 show the questions incorporated in these questionnaires. The first five questions 49 were designed to examine how their perception of water concepts changed after their participation, 50 while the last four served to evaluate how much their attitude towards science had changed. 51 Table 1. Diagnostic evaluation questionnaire of learners' perceptions to science, scientists and
- basic concepts related to water resources.

| Name:                                                                                   | Grade:     |          |          |       |  |  |  |  |  |
|-----------------------------------------------------------------------------------------|------------|----------|----------|-------|--|--|--|--|--|
| Mark with an 'X' the box that better represents your reply.                             |            |          |          |       |  |  |  |  |  |
|                                                                                         | Not at all | A little | Somewhat | A lot |  |  |  |  |  |
| 1 Can you explain the water cycle?                                                      |            |          |          |       |  |  |  |  |  |
| 2 Do you know the meaning of water footprint?                                           |            |          |          |       |  |  |  |  |  |
| 3 How important is to measure precipitation?                                            |            |          |          |       |  |  |  |  |  |
| 4 How much do you know about the instrument to measure rainfall located in your school? |            |          |          |       |  |  |  |  |  |
| 5 Do you know where to consult rainfall data for Mexico City?                           |            |          |          |       |  |  |  |  |  |
| 6 Do you think that science is interesting?                                             |            |          |          |       |  |  |  |  |  |
| 7 Would you like to become a scientist?                                                 |            |          |          |       |  |  |  |  |  |
| 8 Do you agree with this sentence? <u>Science is</u> <u>difficult</u> .                 |            |          |          |       |  |  |  |  |  |
| 9 Do you like to carry out experiments?                                                 |            |          |          |       |  |  |  |  |  |

- Table 2. Final evaluation questionnaire to assess changes in learners' perceptions to science,
- scientists, and basic water resources concepts.

| Name:                                                       | Grade:     |          |          |       |  |  |  |  |
|-------------------------------------------------------------|------------|----------|----------|-------|--|--|--|--|
| Mark with an 'X' the box that better represents your reply. |            |          |          |       |  |  |  |  |
|                                                             | Not at all | A little | Somewhat | A lot |  |  |  |  |
| 1 Do you understand the water cycle better now?             |            |          |          |       |  |  |  |  |
| 2 Can you help reduce the water footprint?                  |            |          |          |       |  |  |  |  |
| 3 How important is to measure precipitation?                |            |          |          |       |  |  |  |  |
| 4 Can you explain how a disdrometer works?                  |            |          |          |       |  |  |  |  |
| 5 Do you know where to consult rainfall data?               |            |          |          |       |  |  |  |  |
| 6 Do you think that science is interesting?                 |            |          |          |       |  |  |  |  |
| 7 Would you like to become a scientist?                     |            |          |          |       |  |  |  |  |
| 8 Do you agree with this sentence? Science is               | 3          |          |          |       |  |  |  |  |
| <u>difficult</u> .                                          |            |          |          |       |  |  |  |  |
| 9 Do you like to carry out experiments?                     |            |          |          |       |  |  |  |  |

The diagnostic questionnaire is applied before the workshop, which allows establishing a baseline 58 of students' perceptions in relation to science and their level of understanding of basic water 59 science concepts. After all activities, the final questionnaire was applied to obtain feedback from 60 the students about the subjects seen. This evaluation instrument helps to infer the students' 61 perceptions towards water science, scientists, and technology before and after its application.

.62

- In summary, the OH-Kids workshop incorporates a series of activities implemented in a successive order (shown in Table 3) along with the time duration and number of students per each activity. The total time for this workshop is 120 minutes per classroom or group of 30 students, considering the application of the evaluation instrument. All activities are developed by the scientific project team; however, teachers are also encouraged to work with the team to encourage
- .68 active pupil participation.
- .69

Table 3. Sequence of OH-Kids workshop activities along with time duration and number of

students.

| Order                       | Activity                                      | Duration<br>(minutes) | Number<br>of learners | Number<br>of groups |
|-----------------------------|-----------------------------------------------|-----------------------|-----------------------|---------------------|
| 1                           | Diagnostic questionnaire                      | 15                    | 30                    | 1                   |
| 2                           | Short introductory talk                       | 15                    | 30                    | 1                   |
| 3                           | Water bingo/memory                            | 15                    | 6                     | 5                   |
| 3                           | Urban water physical model                    | 15                    | 6                     | 5                   |
| 3                           | "Hydro-thon", a water and technology board    | 15                    | 6                     | 5                   |
|                             | game                                          |                       |                       | _                   |
| 3                           | Meet and play with a real optical disdrometer | 15                    | 6                     | 5                   |
| 3                           | Water and technology quiz                     | 15                    | 6                     | 5                   |
| 4                           | Evaluation questionnaire                      | 15                    | 30                    | 1                   |
| Summary per classroom Total |                                               | 120                   | 30                    | 1                   |

The activities workshop starts with a short talk introducing real-time rainfall monitoring at urban

scale and concepts related to water science and technology (i.e. water footprint, hydrological cycle,

precipitation, cloud formation, etc.). This mini talk is aimed at highlighting the importance of

- water for the planet, the cities, and their own lives.
- 77

In the sequence, the group of students is divided in five sub-groups of six students for the application of the activities circuit. This subdivision is carried out to enable the participation of all learners within each activity and to improve teacher-learner relationships, which contributes to a better learning process. In addition, it is acknowledged that low-, medium-, and high-ability students all benefit when being taught in small heterogeneous groups. The learning process of low-ability students may especially suffer risks in homogeneous, teacher-led groups (Wilkinson and Fung 2002).

.85

8

The whole circuit comprises a range of interactive and ludic activities of short duration (15 86 .87 minutes), which are shown as shaded cells in Table 3. These activities are performed 88 simultaneously by each student subgroup, which alternates between different activities every 15 89 minutes. These activities incorporate the work of a facilitator per activity, who supports students 90 who are not willing or able to participate without help. Once all the smaller groups have 91 completed the five activities, the students are regrouped into one plenary session to conclude the 92 workshop and apply the final evaluation questionnaire. Figure 1 illustrates a flow chart of the 93 order of activities during the OH-Kids workshop.

94