# Peer review of "1An innovative STEM outreach model to foster the next generation of2geoscientists, engineers, and technologists (OH-Kids)"

_Geoscience Communication, 2019_

## Referee Comment (RC1) · Anonymous Referee #1 · 20 Jul 2019

1. The problematic approached in the study is actual and has interest concerning the development of 21st century skills in young learners through their involvement in STEM activities.

2. The paper presents an innovative didactic model (OH-Kids) to approach the water cycle and the urban water cycle. The didactic model has many hands-on and minds-on activities. It has also an interesting approach to engage primary school students in science and the scientists work. All major STEM components are present in the activity. However, the activity was not organized in a STEM cycle, but only in a linear and sequencial way. It is also not clear how S&M components are linked in an interdisciplinar

way.

3. The work should present the research question that oriented the study.

The model OH-Kids reached a reasonable amount of 6-12y old students. Though, one major weakness of this work is the questionnaire applied to the students. The questionnaire applied before and after the workshops does not measure the knowledge of the students, but only their perceptions. This is a major limitation of this study and should be referred.

Concerning the didactical sequence of the OH-Kids activities, what was the framework that organized it? I kindly suggest that the authors read: Pedaste, M., Mäeots, M., Siiman, L. A, Jong, T. de, Riesen, S. A. N. van, Kamp, E. T., Manoli, C. C., Zacharia, Z. C., & Tsourlidaki, E. (2015). Phases of inquiry-based learning: definitions and the inquiry cycle. Educational Research Review, 14, 47-61.

4. The results of this study are correct when it is mentioned that this work has the advantage to present a real collaboration between a scientific institution and primary school students. Though, this study lack stronger evidence that the students learn concepts and processes related to the water cycle and other related concepts.

5. The authors gave proper credit to related work.

6. The title is clear; however, I suggest that the expression (OH-kids) should appear this way: "An innovative STEM outreach model (OH-Kids) to foster the next generation of 2 geoscientists, engineers, and technologists

7. The authors mentioned in the abstract that this study was designed to empower educators. However, it's not clear how this study has accomplished that because it is focused in 6-12y students and not in the formation of the educators.

8. The objectives of the OH-Kids model are very clear, however, the goals of the study (68-71) are not clear yet. The focus of the study is on engaging students, teachers, or both in STEM trough the outreach model OH-kids? How the "positive effects" are going

to be measured? That should be express in form of a goal. Concerning the section "2. OH-Kids: a STEM outreach model", I suggest that table 3 and the sequence of OH-Kids should be presented early, after 41-42 lines. These questions should also be clarified: The activities where elaborated by the team or adapted from elsewhere? The activities are related with the science curriculum of Mexico? What is the constitution of the scientific project team that developed the activities? Concerning the section "2.1. Water bingo and memory games", it should be clarified if the cards are organized in themes, as the hydro-thon game. In relation to the section "2.2. Urban water physical model" and figure 4., It could help the interpretation of the figure if the parts of the model were captioned. Add some affirmations of the students concerning their argumentation about causes and consequences of floods could help to present evidences about their interest about science. About the section "2.5 Water and technology quiz", what application was used to deliver the quiz? The quiz was answered by each student individually or in small groups?

9. The language is clear and precise.

10. The work presents in p.5 a reference to an OECD study from 2016. This study was not in the references, but only "OCDE (2015)". The authors use a quite old work of Bacon (1987) to approach gamification. There are many other contemporary works about this issue that could be addressed.

---

## Referee Comment (RC2) · Anonymous Referee #2 · 1 Aug 2019

O texto do artigo apresenta uma série de atividades realizadas com crianças na faixa etária de 6-12 anos, abordando a temática da água e seu contexto urbano. Considera atividades lúdicas apropriadas para a faixa etária e aborda conteúdos específicos relacionados com o cotidiano das crianças. O texto é claro, as descrições das atividades são de fácil compreensão. Há uma avaliação da percepção das crianças participantes do projeto sobre os conhecimentos abordados e sobre o interesse pela ciência. O tema abordado no artigo é relevante e contextualizado com a realidade dos alunos. O projeto apresenta objetivos claros e bons resultados. Os métodos de ensino envolvem e despertam o interesse dos participantes, sendo apropriados para a faixa etária. Na minha opinião, a discussão dos resultados deve mostrar de forma clara que

a avaliação é sobre a percepção dos alunos e não sobre a aprendizagem. Para uma avaliação da aprendizagem, os proponentes do projeto poderiam envolver os professores das escolas, numa relação contínua com o currículo escolar e com outros temas que os professores trabalham em sala de aula. A relação de continuidade com o currículo favorece o aprendizado das crianças e contribui para que a atividade não seja apenas pontual, durante o tempo do workshop. Isso envolveria também um trabalho de formação dos professores, que passariam a ser integrantes do projeto e não mero expectadores. O artigo está bem escrito e apresenta uma experiência importante de disseminação dos conhecimentos das geociências nas escolas.

---

## Author Comment (AC1) · 26 Sep 2019

We warmly thank the anonymous referee 1 for the time spent during the thorough revision of the article. Also, we would like to thank the referee 1 for the depth and detailed feedback and constructive comments on the study. We believe that the incorporation of the suggestions and recommendations from referee 1 will result in a better version of the article.

In the following lines, we explain our reply with an "R:" after the "Referee comment". Text that we suggest can change or be added to the article appear in underlined italics and the paragraph in quotes, according with recommendations of referee.

[Figure]

—————————————point by point response—————————————

"Referee comment 1": 1. The problematic approached in the study is actual and has interest concerning the development of 21st century skills in young learners through their involvement in STEM activities.

"Response 1": First, we warmly thank the reviewer for this opinion. This is the reason that triggered our interest and work on the topic. We are convinced of the necessity to make available results from research projects to the younger members of society.

"Referee comment 2": 2. The paper presents an innovative didactic model (OH-Kids) to approach the water cycle and the urban water cycle. The didactic model has many hands-on and minds-on activities. It has also an interesting approach to engage primary school students in science and the scientists work. All major STEM components are present in the activity. However, the activity was not organized in a STEM cycle, but only in a linear and sequencial way. It is also not clear how SM components are linked in an interdisciplinary way.

"Response 2": The comment about the organization of activity cycle will be answered in the "Response 3-3". About the SM components, we have given a great effort on explaining that mathematical concepts are linked with water science in an interdisciplinary context during OH-Kids activities. For example, when we tackle the concept of measuring precipitation we teach the units of measurement and unit conversion of length, area and volume, with the objective that pupils comprehend the significance of large rainfall quantities in an urban area. Also, in the disdrometer activity the rain data is displayed as graphs and the pupils learn how to interpret these results and the importance that this is within the water science discipline. In order to incorporate a better explanation of the integration of SM components in the activities, we suggest some changes and new phrases in the paragraph L 173 and lines L 325-326 and L 330:

L 173:
"The activities workshop starts with a short talk introducing a real-time rainfall monitoring *system* at urban scale and concepts related to water science (i.e. water footprint, hydrological cycle, cloud *and precipitation* formation, etc.), *highlighting the integration of mathematics and technology concepts (i.e. instrumentation and measurement of precipitation, etc.).* This mini talk accentuates the importance of water for the planet, the cities, and their own lives. *One important topic that is addressed are units of measurement and unit conversion with the objective that pupils comprehend the significance of large rainfall quantities in an urban area.*"

L 325-326:

"The *numerical* information is immediately displayed through graphs in a screen minute by minute."

L 330 (before "...spray bottle." And after "After the..."):

"*The OH team encourage pupils to interpret the information measured and displayed as graphs, therefore obtaining the sensibility to distinguish the difference between small and large rainfall quantities and that the rain drops don't have the same size.*"

"Referee comment 3": (here we separate the comments in three parts (identified by "3-X") to better comprehend)

"3-1": The work should present the research question that oriented the study.

"Response 3-1": Thank you for your comments and suggestions. With regards to the research question, this study was motivated by the following question: "Is it possible to enhance learning experiences of STEM disciplines for students (age 6-12), using a water resource research project and different pedagogical strategies?". In order to attend the recommendation, we will add a new phrase in the initial of paragraph L 68 that talks about the objective (introduction):

L 68 (before "This study..."):

"*Within this context, the research question that motivates this study is: "Is it possible to enhance learning experiences of STEM disciplines for students (age 6-12), using a water resource research project and different pedagogical strategies?".*"

"3-2": The model OH-Kids reached a reasonable amount of 6-12y old students. Though, one major weakness of this work is the questionnaire applied to the students. The questionnaire applied before and after the workshops does not measure the knowledge of the students, but only their perceptions. This is a major limitation of this study and should be referred.

"Response 3-2": Regarding the questionnaire, we agree with your comment, this is a major limitation of our study. Really, the questionnaire developed and applied during the workshop doesn't permit us to verify the knowledge of the pupils. This point already has been discussed by our team and for future workshops we are considering on incorporating other methodology of evaluation. To emphasize this, we suggest to reorganize the paragraphs between L 476-482 and add a new phrase:

L 476-482:

"The results from the OH-Kids workshop in a sample of 344 students (6-12 year-olds) show that the activities and applied techniques of the workshop provided apparent changes in the initial perceptions of some students about concepts and ideas *referring to* science and scientists. Furthermore, the results suggest that including different ways of communicating *concepts* related to STEM disciplines broadens students' interest and motivation. This was shown by the comparison of answers between the diagnostic and final evaluation questionnaires. *It's important to emphasize that one of the limitations of this study is related to the workshops method of evaluation, because it does not allow to measure changes in the knowledge about the themes covered.*"

"3-3": Concerning the didactical sequence of the OH-Kids activities, what was the framework that organized it? I kindly suggest that the authors read: Pedaste, M.,

Mäeots, M., Siiman, L. A, Jong, T. de, Riesen, S. A. N. van, Kamp, E. T., Manoli, C. C., Zacharia, Z. C., Tsourlidaki, E. (2015). Phases of inquiry-based learning: definitions and the inquiry cycle. Educational Research Review, 14, 47-61.

"Response 3-3": Concerning the sequence of the activities, the circuit of OH-Kids model activities is based on a constructivist pedagogy, sociocultural theory of Vygotsky and principles of different learning strategies, for example discovery, experiential, game-based learning; offering pupils a multi-sensory learning experience. In this context, our model is not organized in one specific learning cycle for STEM, for example Kolb's cycle or inquiry cycle; but instead we used fragments of the learning strategies mentioned above. The OH-Kids model is a didactic strategy for outreach of a project which involves STEM disciplines in a real-life problem: rainfall monitoring in urban area, that is, the students are not instructed to solve a real-life problem. In order to make this more explicit we will add more details in the paragraph L 97:

L 97:

"The OH-Kids outreach model comprises a collaborative and effective researcher-educator-learner communication strategy - built around *constructivist pedagogy, sociocultural theory and principles of engagement, guided discovery, experiential and game-based learning* - to encourage STEM subject uptake."

"Referee comment 4": 4. The results of this study are correct when it is mentioned that this work has the advantage to present a real collaboration between a scientific institution and primary school students. Though, this study lack stronger evidence that the students learn concepts and processes related to the water cycle and other related concepts.

"Response 4": How we mentioned in the "Response 3", unfortunately, the evaluation method used in this study does not provide evidence that students really learned the concepts, but only indicate changes in perceptions about these topics.
"Referee comment 5": 5. The authors gave proper credit to related work.

"Response 5": Thank you for your comment.

"Referee comment 6": 6. The title is clear; however, I suggest that the expression (OH-kids) should appear this way: "An innovative STEM outreach model (OH-Kids) to foster the next generation of 2 geoscientists, engineers, and technologists"

"Response 6": We agree with your recommendation. In the new version of this we have edited the title as suggested.

"Referee comment 7": 7. The authors mentioned in the abstract that this study was designed to empower educators. However, it's not clear how this study has accomplished that because it is focused in 6-12y students and not in the formation of the educators.

"Response 7": We agree that this study does not focus on the formation of the educators. When we mentioned 'empower educators', the intention was to awaken them to new strategies for teaching STEM disciplines, since they are spectators of the activities, how we mentioned in L 127-128. In this sense, we considered important to remove the words "educators and" in the text lines L 14 and L 70:

L 13-15:

"To address this need, an outreach model entitled OH-Kids was designed to *empower pupils* through the development of high-quality STEM learning experiences based on a research project."

L 70:

". . . *to empower pupils* through the development of STEM learning experiences focused on water resources and based on a research project."

"Referee comment 8": (here we separate the comments in three parts (identified by "8-X") to better comprehend)

"8-1": The objectives of the OH-Kids model are very clear, however, the goals of the study (68-71) are not clear yet. The focus of the study is on engaging students, teachers, or both in STEM trough the outreach model OH-kids? How the "positive effects" are going to be measured? That should be express in form of a goal.

"Response 8-1": We would like to warmly thank the referee for the observations about the objectives and activities. Following the referee's comments, we suggest to edit the objectives of this study that appear in paragraph L 68 (remember that following your comment 3 we proposed to include a research question in line L 68):

L 68:

"*Therefore, this study demonstrates an innovative teaching strategy via the outreach model entitled OH-Kids created* to empower pupils through the development of STEM learning experiences focused on water resources and based on a research project. *Furthermore, the study presents the results of its application in schools of Mexico City.*"

"8-2": Concerning the section "2. OH-Kids: a STEM outreach model", I suggest that table 3 and the sequence of OH-Kids should be presented early, after 41-42 lines.

"Response 8-2": With regards to the location of Table 3 and Figure 1 in item 2, we agree with your suggestion. In order to attend it, we suggest that the Table 3 and Figure 1 are moved to line 143 and some changes in text: add "(Table 3)" in line L 137, move the phrase "Figure 1 illustrates. . ." of the line L 192 to L 142 and add "(as shown in Figure 1)" in line L 189. Here we show the paragraphs L 137 and L 186 with these changes:

L 137:

"The model is akin to a workshop and includes two types of activities *(Table 3)*. The first one is related to traditional academic strategies: a short lecture that explains the real-world problem and associated concepts. The second one consists of interactive and ludic activities that are implemented within the classroom as didactic tools for teaching the STEM disciplines involved in water re­sources. All activities are organised in a circuit that enables the alternation of an academic activity with one of a more ludic nature, which can be considered as one of the innovations of this outreach model. *Figure 1 illustrates a flow chart of the order of activities during the OH-Kids workshop.*"

L 186:

"The whole circuit comprises a range of interactive and ludic activities of short dura­tion (15 minutes), which are shown as shaded cells in Table 3. These activities are performed simultaneously by each student subgroup, which alternates between differ­ent activities every 15 minutes *(as shown in Figure 1)*. These activities incorporate the work of a facilitator per activity, who supports students who are not willing or able to participate without help. Once all the smaller groups have completed the five activities, the students are regrouped into one plenary session to conclude the workshop and apply the final evaluation questionnaire."

"8-3": These questions should also be clarified: The activities where elaborated by the team or adapted from elsewhere? What is the constitution of the scientific project team that developed the activities?

"Response 8-3": Some activities were elaborated by team members (meet a real dis­drometer and urban physic model) but others were just adapted (bingo, memory game, Hidro-thon) to teach water science, engineer and technology topics. Regarding the constitution of the scientific project team, consist of a multidisciplinary team with civil, electronic and environmental engineers, biology, physics and educators; making the activities to have a systemic view. To clarify the question about the creation/adaptation of activities and scientific project team we will change in the text lines L 166-168 and L 202-206:

L 163-168:

"*The activities are applied by the multidisciplinary scientific project team that comprise civil, electronic and environmental engineers, biology, physics and educators*; however, teachers are also encouraged to work with the team to encourage active pupil participation."

L 202-206:

"The five activities of the circuit were *developed* considering visual and tactile stimuli, as well as observation, attention, memory and concentration. In addition, they were put together to reinforce the concepts introduced in the initial talk and to spark interest in STEM disciplines, using evidence and demonstrations from the research project as a basis (Renninger and Su, 2012). *It's important to mention that some activities were adapted (water bingo, memory games, Hydro-thon) and others elaborated (meet a real optical disdrometer, urban water physical model) by team.*"

"8-4": The activities are related with the science curriculum of Mexico?

"Response 8-4": Regarding the science curriculum of Mexico, the public education program mentions 'natural science and technology', as part of the primary education curriculum and where it includes themes about the water cycle. However, in general, schools teach only basics concepts without a current overview and relationship with STEM. In this sense, the content of all activities of the OH-Kids model are related with the science curriculum of Mexico and additionally we include more advanced concepts of water science linked to STEM. In Mexico and Latin America, the topics of STEM require a higher positioning within the educational curriculum to enhance positive effects on the attitude of students towards these disciplines. This is the current challenge. Although, recently the Mexican public education program has stimulated the use of principles of the scientific method for certain activities, for example the exploration. This scenario is one of the reasons that motivated us to bring new activities for primary education.

"8-5": Concerning the section "2.1. Water bingo and memory games", it should be clarified if the cards are organized in themes, as the hydro-thon game.

"Response 8-5": To attend your suggestion about the themes of the water bingo and memory games, we suggest to edit the phrases in L 228-232:

L 228-232 (after "…for the game.")

"*Every image represents a term in relation to one of following topics: 1. concepts of hydrology (i.e.: water cycle, hyetograph, type of precipitation), 2. meteorology concepts (i.e.: hurricane, floods, climate change), 3. technological terms (i.e.: solar panel, weather radar, raspberry pi) and 4. names of OH stations.*"

"8-6": In relation to the section "2.2. Urban water physical model" and figure 4., It could help the interpretation of the figure if the parts of the model were captioned.

"Response 8-6": Additionally, as suggested we propose to include more details about the parts of the physical model in the title of the Figure 4 L 272 and specify the corresponding parts of the Figure 4 in the text L 260:

L 272:

"Figure 4. Details of the urban water physical model constructed for the OH-Kids workshop *(a – glass case with representation of small-scaled city; b – urban drainage system and c – rainfall simulator).*"

L 260 (here we show the complete phrase L 258-260):

"Our physical model consists of a small-scaled city within a glass case, a miniature urban drainage, and rainfall simulator *(Figure 4 – a, b and c, respectively).*"

"8-7": Add some affirmations of the students concerning their argumentation about causes and consequences of floods could help to present evidences about their interest about science.

"Response 8-7": With regards to the students' interest about science during the physical model activity, we have made an effort to provide a better evidence. In order to attend your suggestion we will add a new phrase in L 280:

L 280 (after "...and justification (Sheldrake et al., 2017)." and before "The response..."):

"*The pupils were impressed to find out what a drainage system is, how it works and how important it is, they also argued a lot about the importance of throwing the garbage in the right place and the danger of walking on the street when rain is very strong.*"

8-8": About the section "2.5 Water and technology quiz", what application was used to deliver the quiz? The quiz was answered by each student individually or in small groups?

"Response 8-8": The last point is associated with the application used to deliver the quiz and the number of participants. As pointed out there is a lack of information and in order to include it we propose to add new information in the lines L 368-372:

L 368-372

"This game is displayed in an interactive monitor *through a presentation editor software,* and consist of ten questions about water and technology. Players need to answer the quiz sequentially and a point is given for each correct answer. *The quiz can be played individually or in pairs (as we apply).* Considering the learning process, this game replicates an exam-like situation, but in an interactive setting. Figure 9 shows a group of students playing the quiz game during one of our workshops."

"Referee comment 9": 9. The language is clear and precise.

"Response 9": Thank you very much for your comment.

"Referee comment 10":10. The work presents in p.5 a reference to an OECD study

from 2016. This study was not in the references, but only "OCDE (2015)". The authors use a quite old work of Bacon (1987) to approach gamification. There are many other contemporary works about this issue that could be addressed.

"Response 10": Thanks for pointing this out. We checked the OECD reference and the title corresponds to 'PISA 2015 results (Volume I)'. Following the references format used in the manuscript, the publish year appears at the end of the phrase (in this case 2016). In order to attend your recommendation about the contemporary reference, we suggest incorporating another author that speaks about the same topic (Bochennek et al., (2007)) and for it we propose to rephrase the sentence in L 365-368 as follows:

L 365-368

"*In agreement with Bochennek et al., (2007) games could be seen as experiential learning cycles in that they repeat learning stages in each game turn or every game played. In this context, the authors mention that games can be classified in various model stages according to the different learning processes involved. The quiz developed for our workshop may be categorised as a two-stage model (Bochennek et al. (2007) based on Bacon (1987)), where experience is followed by reflection.*"

"*Bochennek, K., Wittekindt, B., Zimmermann, S. Y., Klingebiel, T.: More than mere games: a review of card and board games for medical education, Medical Teacher, 29:9-10, 941-948, DOI: 10.1080/01421590701749813, 2007.*"

---

## Author Comment (AC2) · 26 Sep 2019

We would like to thank anonymous referee 2 for reviewing the manuscript and providing us with his kind words and constructive comments. We have responded to the general comments as outlined below. Text that we suggest change or add in the article appear in underlined italics and the paragraph in quotes.

Referee2 general comment:

O texto do artigo apresenta uma série de atividades realizadas com crianças na faixa

etária de 6-12 anos, abordando a temática da água e seu contexto urbano. Considera atividades lúdicas apropriadas para a faixa etária e aborda conteúdos específicos relacionados com o cotidiano das crianças. O texto é claro, as descrições das atividades são de fácil compreensão. Há uma avaliação da percepção das crianças participantes do projeto sobre os conhecimentos abordados e sobre o interesse pela ciência. O tema abordado no artigo é relevante e contextualizado com a realidade dos alunos. O projeto apresenta objetivos claros e bons resultados. Os métodos de ensino envolvem e despertam o interesse dos participantes, sendo apropriados para a faixa etária. Na minha opinião, a discussão dos resultados deve mostrar de forma clara que a avaliação é sobre a percepção dos alunos e não sobre a aprendizagem. Para uma avaliação da aprendizagem, os proponentes do projeto poderiam envolver os professores das escolas, numa relação contínua com o currículo escolar e com outros temas que os professores trabalham em sala de aula. A relação de continuidade com o currículo favorece o aprendizado das crianças e contribui para que a atividade não seja apenas pontual, durante o tempo do workshop. Isso envolveria também um trabalho de formação dos professores, que passariam a ser integrantes do projeto e não mero expectadores. O artigo está bem escrito e apresenta uma experiência importante de disseminação dos conhecimentos das geociências nas escolas.

Response:

Primeiramente, gostaríamos de agradecer ao referee2 pelo tempo dedicado para a revisão completa do nosso artigo. Seu feedback foi muito importante para nós, porque mostrou que conseguimos transmitir todas as informações relevantes do nosso projeto. Seus comentários construtivos e suas recomendações ajudaram a melhorar o artigo e serão muito importantes para o futuro do projeto. Em relação a avaliação do modelo OH-Kids, sempre buscamos mencionar em todo o texto que as atividades aplicadas provocaram mudanças nas percepções dos alunos aos temas trabalhados, dado que foi o objetivo neste momento. Entretanto, seguindo a sua recomendação de mostrar de forma clara que a avaliação é sobre a percepção dos alunos e não a

aprendizagem, bem como a sugestão do referee1, propomos a reestruturação do texto entre as linhas L 476-482 e a inclusão de uma nova frase aonde ressaltamos que esta es uma das limitaçes do estudo:

L 476-482:

"The results from the OH-Kids workshop in a sample of 344 students (6-12 year-olds) show that the activities and applied techniques of the workshop provided apparent changes in the initial perceptions of some students about concepts and ideas *referring to* science and scientists. Furthermore, the results suggest that including different ways of communicating *concepts* related to STEM disciplines broadens students' interest and motivation. This was shown by the comparison of answers between the diagnostic and final evaluation questionnaires. *It's important to emphasize that one of the limitations of this study is related to the workshops method of evaluation, because it does not allow to measure changes in the knowledge about the themes covered.*"

Para futuras aplicações do modelo OH-Kids nosso grupo considerará o uso de novas metodologias de avaliação para poder analisar as implicações na aprendizagem dos alunos. Neste sentido, suas sugestões serão tomadas em conta para a nossa decisão. Também acreditamos que o processo de avalição da aprendizagem e do alcance das atividades do workshop deve considerar a participação dos professores, uma vez que eles têm maior contato e conhecimento da evolução dos alunos. Devido as nossas experiências, o tema da formação dos professores sempre foi discutido pelo nosso grupo de pesquisa, uma vez que notamos a necessidade de compartilhar conhecimento técnico sobre a importância das disciplinas de STEM para soluções da vida cotidiana em diferentes áreas. Além disso, para estimular o desenvolvimento de atividades inovadoras para despertar o interesse dos alunos aos temas. De esta forma, este tema será tratado com importância para a extensão do projeto. Para finalizar, novamente agradecemos todos os comentários do referee 2 e ficamos muito felizes em saber que o tema deste artigo é de seu interesse e que o considera relevante.